# Regulation of Th17/Treg Balance by 27-Hydroxycholesterol and 24S-Hydroxycholesterol Correlates with Learning and Memory Ability in Mice

**DOI:** 10.3390/ijms23084370

**Published:** 2022-04-15

**Authors:** Tao Wang, Shanshan Cui, Ling Hao, Wen Liu, Lijing Wang, Mengwei Ju, Wenjing Feng, Rong Xiao

**Affiliations:** Beijing Key Laboratory of Environmental Toxicology, School of Public Health, Capital Medical University, Beijing 100069, China; wangtao_930106@163.com (T.W.); cuishanshan@ccmu.edu.cn (S.C.); hl_aoing@163.com (L.H.); aaa200841@163.com (W.L.); w1312180@163.com (L.W.); meave_ju@163.com (M.J.); 15810888862@163.com (W.F.)

**Keywords:** 27-hydroxycholesterol, 24S-hydroxycholesterol, RORγt, Th17/Treg, learning and memory ability

## Abstract

Dysregulation of cholesterol metabolism and its oxidative products—oxysterols—in the brain is known to be associated with neurodegenerative diseases. It is well-known that 27-hydroxycholesterol (27-OHC) and 24S-hydroxycholesterol (24S-OHC) are the main oxysterols contributing to the pathogenesis of Alzheimer’s disease (AD). However, the molecular mechanism of how 27-OHC and 24S-OHC cause cognitive decline remains unclear. To verify whether 27-OHC and 24S-OHC affect learning and memory by regulating immune responses, C57BL/6J mice were subcutaneously injected with saline, 27-OHC, 24S-OHC, 27-OHC+24S-OHC for 21 days. The oxysterols level and expression level of related metabolic enzymes, as well as the immunomodulatory factors were measured. Our results indicated that 27-OHC-treated mice showed worse learning and memory ability and higher immune responses, but lower expression level of interleukin-10 (IL-10) and interferon (IFN-λ2) compared with saline-treated mice, while 24S-OHC mice performed better in the Morris water maze test than control mice. No obvious morphological lesion was observed in these 24S-OHC-treated mice. Moreover, the expression level of interleukin-17A (IL-17A), granulocyte-macrophage colony-stimulating factor (GM-CSF) and macrophage inflammatory protein 3α (MIP-3α) were significantly decreased after 24S-OHC treatment. Notably, compared with 27-OHC group, mice treated with 27-OHC+24S-OHC showed higher brain 24S-OHC level, accompanied by increased CYP46A1 expression level while decreased CYP7B1, retinoic acid-related orphan receptor gamma t (RORγt) and IL-17A expression level. In conclusion, our study indicated that 27-OHC is involved in regulating the expression of RORγt, disturbing Th17/Treg balance-related immune responses which may be associated with the learning and memory impairment in mice. In contrast, 24S-OHC is neuroprotective and attenuates the neurotoxicity of 27-OHC.

## 1. Introduction

Alzheimer’s disease (AD) is a debilitating neurodegenerative disease characterized by the progressive impairment of learning and memory ability [1]. The typical neuropathological and diagnostic markers of AD are the aggregation of β-amyloid (Aβ) plaques and phosphorylated tau forming neurofibrillary tangles (NFTs) [2].

Maintenance of cholesterol homeostasis in the brain is critical for neuronal transmission, synapse formation and brain development [3]. More and more evidence indicates that abnormal cholesterol metabolism participates in the pathogenesis of AD [4]. The level of cholesterol in the brain is largely independent of its peripheral level due to the blood brain barrier (BBB). Recently, epidemiological and experimental studies have shown that cholesterol influences nerve function through its oxidized derivatives—oxysterol—which can cross the BBB at a much faster rate and regulate neuron survival, neural development, brain aging and neurodegeneration [5,6]. In neurons, the brain-specific enzyme CYP46A1 catalyzes the production of 24S-hydroxycholesterol (24S-OHC) in response to the excessive cholesterol accumulation, which diffuses from the brain into the circulation for degradation. Simultaneously, 27-hydroxycholesterol (27-OHC), a peripheral metabolite of cholesterol catalyzed by CYP27A1, flows into the brain [7]. Studies show that 27-OHC is a risk factor for developing and aggravating AD pathology by disrupting cholesterol metabolism, reducing glucose uptake, activating the renin–angiotensin system (RAS) and increasing Aβ and phosphorylated tau protein in the brain, while 24S-OHC can be a protective factor in AD. Lower levels of 24S-OHC were detected in all brain areas of AD patients [8,9,10]. The interplay between 27-OHC and 24S-OHC may be a bridge between cholesterol metabolism dysfunction and learning and memory ability, but the potential mechanism is still unclear.

Evidence supports the roles of oxysterols in the progression of diseases by binding to and modulating the activity of nuclear receptors, such as liver X receptor (LXR), estrogen receptor (ER), retinoic acid-related orphan receptor (ROR) and the glucocorticoid receptor (GR), etc. [11]. Given the interaction between oxysterols and RORγt, which is specifically involved in T helper 17 (Th17) cells differentiation and functionality, studies of oxysterols on AD progression by disrupting the immune responses are gaining increasing attention over the last decades [12].

The differentiation of naive T cells into Th17 and regulatory T (Treg) cells requires unique transcription factors RORγt and forkhead box protein p3 (Foxp3), respectively, which share common signaling pathways mediated by transforming growth factor β (TGF-β) [13]. It was reported that the differentiation and activities of Th17 cells were increased in AD patients [14,15]. Treg cells are a special subset of immunosuppressive T cells, acting as critical negative regulators of immunity in various pathological states including AD [16]. Therefore, the balance between Th17 and Treg cells as well as the related transcription factors and cytokines is crucial. Previous studies showed that 27-OHC is an RORγt agonist while 24S-OHC is an inverse agonist. Changes in oxysterol homeostasis could either promote or inhibit the transcriptional activity of RORγt [17,18]. The activation of RORγt and production of key cytokine—interleukin-17A (IL-17A)—induced Aβ deposition, neuroinflammation, microglia activation and increased neutrophil counts, all of which were the major physiopathologic mechanisms of AD [19]. Moreover, Dr. Lee suggested that serum amyloid A proteins (SAA), the protein precursor of reactive AA amyloidosis, had distinct local and systemic functions in promoting Th17-mediated immune diseases [20]. Sano et al. also reported that SAA acted directly on the Th17 cells by enhancing their differentiation and effector functions, contributing to inflammatory diseases through the interleukin-23 receptor (IL-23R)/interleukin-22 (IL-22) circuit. SAA oligomers disrupt the cellular membrane and release the amyloid into the extracellular space, resulting in extensive deposition of extracellular AA amyloid [21], which is similar to the manner of Aβ in the pathogenesis of AD [22,23]. Consequently, there seems to be some sort of association between SAA and Th17-mediated immune dysfunction, while the underlying mechanisms remain largely unaddressed.

In this study, we used different oxysterol-treated C57BL/6J mice to explore the possible mechanisms associated with 27-OHC- and 24S-OHC-induced immune dysregulation and neurotoxic effects. We propose that 27-OHC and 24S-OHC disturb the level of oxysterols and the expression of their metabolic enzymes, coupled with an impact on Th17/Treg balance-related immune responses, and eventually morphological lesions in the brain and changes of learning and memory ability.

## 2. Results

### 2.1. Effects of Oxysterols on Body Weight and Organ Coefficient

In this study, C57BL/6J mice were subcutaneously injected with 27-OHC, 24S-OHC or 27-OHC+24S-OHC for 21 days, followed by novel object recognition and Morris water maze test. After the behavioral experiments, we measured the body weight and the organ weight of all animals to evaluate the differences of treatments on the growth and organ weight-related effects. As shown in Figure 1A, there was no significant difference of body weight before and after three weeks’ treatment (*p* > 0.05). Moreover, we also calculated the organ coefficient (organ weight/body weight, %). As shown in Figure 1D, the heart coefficient in the 27-OHC group showed significant changes compared with 24S-OHC group (*p* = 0.038) and 27-OHC+24S-OHC group (*p* = 0.017), while no significant change was found in the spleen, liver and kidney coefficient in the four groups (*p* > 0.05) (Figure 1B,C,E).

### 2.2. Oxysterols Treatment Significantly Affected Mice Learning and Memory Ability

The novel object recognition and Morris water maze test were carried out to evaluate the effects of oxysterols on learning and memory ability of mice. In novel object recognition test, the novel object recognition index (NORI), frequency discrimination index (FDI) and time discrimination index (TDI) were calculated to determine the short-term/working memory of mice. There was no significant difference in NORI, FDI and TDI values (*p* > 0.05) (Table 1). However, as shown in Figure 2A,B, 27-OHC-treated mice showed a decreasing trend in the NORI and FDI value, while the 24S-OHC-treated group exhibited an increasing trend.

The Morris water maze test was followed to evaluate the spatial learning and memory of mice. During the orientation navigation tests, there were remarkable differences of escape latency on Day5 (F = 4.844, *p* = 0.001) and the mean distance to platform (F = 3.022, *p* = 0.047). The escape latency in 24S-OHC group on Day5 was shorter than Day1 (*p* = 0.024) and Day2 (*p* = 0.045), suggesting better spatial learning ability of mice was obtained by repeated trials (Figure 2E). Significantly increased numbers of platform-site crossovers were also shown in the 24S-OHC group compared with the 27-OHC (*p* = 0.006) and 27-OHC+24S-OHC group (*p* = 0.013) (Figure 2H). The 27-OHC-treated mice showed significantly increased distance to the platform when compared to the Control (*p* = 0.017) and 24S-OHC (*p* = 0.017) group (Figure 2F). These results were supported by representative images of swimming path during the probe trails (Figure 2D). No significant difference was observed in average speed and the time/distance spent in the target quadrant (*p* > 0.05) (Figure 2G,I,J).

### 2.3. Oxysterols Treatment Significantly Affected the Brain Pathology

To study whether oxysterol treatment affects the brain histopathology, HE staining was carried out and presented in Figure 3A,C. In the 27-OHC treatment group, the number of neurons in the hippocampal CA1 region was significantly reduced compared with the Control group (*p* = 0.029) and the cells were irregular and disordered in shape and size. Nuclei pyknosis (black arrow) and enlarged intercellular space were observed to varying degrees. In comparison, the number of neurons in the 24S-OHC group was significantly higher than that of 27-OHC (*p* = 0.013) and 27-OHC+24S-OHC (*p* = 0.026) group. The hippocampal neurons of 24S-OHC-treated mice were neatly arranged and structurally intact, and the membrane of cells and nucleus were clear.

Given that APP and SAA are critical amyloid precursors involved in amyloidosis, their mRNA and protein expressions in the brain were detected in our study. Compared with the Control group, the expression of the APP (gene: *p* = 0.006) and SAA (gene: *p* < 0.001, protein: *p* = 0.036) were upregulated in the 27-OHC group, while the results of APP protein and SAA mRNA level in the 24S-OHC group were down-regulated. The expression of APP (protein: *p* < 0.001) and SAA (gene: *p* = 0.017, protein: *p* = 0.035) in the 27-OHC group was also higher than that of 24S-OHC mice. Notably, APP protein expression in the 27-OHC+24S-OHC group was upregulated compared with 24S-OHC (*p* = 0.006), but the SAA protein was downregulated in the 27-OHC group (*p* = 0.014) (Figure 3D–H).

### 2.4. Changes of Serum/Brain Oxysterols and the Expression of Metabolic Enzymes in Brain

The level of oxysterols in the serum and brain was detected by high-performance liquid chromatography-mass spectrometry (HPLC-MS). As shown in Figure 4A,B, 27-OHC treatment markedly increased the level of 27-OHC in serum compared with Control (*p* = 0.006) and 24S-OHC (*p* = 0.014). The serum 27-OHC level in 27-OHC+24S-OHC mice was also higher than the Control (*p* = 0.040) group. There was no significant change of brain 27-OHC level after the treatment (F = 1.069, *p* = 0.415). Furthermore, compared with the 27-OHC group, the serum 24S-OHC in the Control group (*p* = 0.034), brain 24S-OHC level in the 24S-OHC (*p* = 0.012) and 27-OHC+24S-OHC (*p* = 0.025) group were significantly increased (Figure 4C,D).

We also measured the mRNA and protein expression level of oxysterol-related metabolic enzymes in the brain (Figure 4E–K). CYP27A1 and CYP46A1 are the synthetases of 27-OHC and 24S-OHC, respectively. CYP7B1 is responsible for the catabolism of 27-OHC [24]. It was shown that the CYP27A1 (*p* = 0.008) and CYP7B1 (*p* = 0.001) protein level was significantly upregulated in the 27-OHC group compared to the Control group, while CYP46A1 was downregulated (gene: *p* = 0.017, protein: *p* = 0.003). Both the gene and protein expression of CYP27A1 (gene: *p* = 0.007, protein: *p* = 0.007) and CYP7B1 (gene: *p* = 0.037, protein: *p* < 0.001) in the 27-OHC group were evidently enhanced compared to the 24S-OHC group, whereas CYP46A1 was down-regulated (gene: *p* = 0.012, protein: *p* = 0.001). Moreover, the expression of CYP27A1 mRNA was visibly increased in the 27-OHC+24S-OHC group compared to the 24S-OHC (*p* = 0.007). Similarly, contrary to the 27-OHC+24S-OHC group, increased CYP7B1 protein (*p* < 0.001) but decreased CYP46A1 protein (*p* = 0.001) were found in the 27-OHC group.

### 2.5. Oxysterols Treatment Significantly Affected the Expression Level of T-Cell-Specific Transcription Factors and the Immunomodulatory Factors in the Brain

To evaluate the effect of oxysterols on the specific transcription factors of Th17 and Treg cells, we detected the expression of RORγt and Foxp3 in the brain of mice [14,25]. As shown in Figure 5A,C, the mRNA and protein expression of RORγt in the 27-OHC group were significantly upregulated compared with the Control group (gene: *p* < 0.001; protein: *p* = 0.015) and 24S-OHC (gene: *p* < 0.001; protein: *p* = 0.002) group. Its mRNA expression in the 27-OHC group was also higher than the 27-OHC+24S-OHC (*p* = 0.027) group. Moreover, compared to the 27-OHC+24S-OHC group, the mRNA and protein expression of RORγt were significantly downregulated in the 24S-OHC (gene: *p* < 0.001; protein: *p* = 0.025) group. In contrast, the expression of the Foxp3 gene in the 24S-OHC group was increased compared with the other three groups (Control: *p* = 0.001; 27-OHC: *p* < 0.001; 27-OHC+24S-OHC: *p* = 0.002). The expression of Foxp3 protein in the 24S-OHC group was also increased, but the significant differences were only observed in the Control group (*p* = 0.040) (Figure 5B,D).

Furthermore, to examine the effect of oxysterols on the key immune factors in Th17/Treg balance, the expression of IL-17A, granulocyte-macrophage colony-stimulating factor (GM-CSF), macrophage inflammatory protein 3α (MIP-3α), interleukin-10 (IL-10) and interferon-λ2 (IFN-λ2) in the brain were determined in this study (Figure 5E–O). Compared with the 24S-OHC group, the expression of IL-17A (gene: *p* < 0.001; protein: *p* = 0.015), MIP-3α (mRNA: *p* < 0.001) and GM-CSF (gene: *p* = 0.018; protein: *p* = 0.006) was significantly upregulated in 27-OHC mice. Meanwhile, the expression of the factors was also higher in the Control group than the 24S-OHC group (MIP-3α gene: *p* = 0.001, IL-17A gene: *p* = 0.001; IL-17A protein: *p* = 0.001, GM-CSF protein: *p* = 0.017). Besides, the expression of IL-17A in the 27-OHC group was increased compared to the Control (gene: *p* = 0.003) and 27-OHC+24S-OHC (gene: *p* < 0.001; protein: *p* = 0.013) group. While in the 24S-OHC group, it was significantly decreased compared to those of the Control (gene: *p* = 0.001) and 27-OHC+24S-OHC group (gene: *p* = 0.028, protein: *p* = 0.042). Similarly, the expression of the MIP-3α gene in the 24S-OHC group was significantly lower than the 27-OHC+24S-OHC (*p* = 0.001) group. With regard to immunosuppressive factors, the expression of IL-10 (gene: *p* = 0.017; protein: *p* = 0.040) and IFN-λ2 (gene: *p* = 0.001) in the 24S-OHC group was higher than the 27-OHC group. However, no significance was found in change of the protein expression level of MIP-3α and IFN-λ2.

## 3. Discussion

Oxysterol homeostasis is crucial for cholesterol metabolism in the brain [26]. Evidence suggests the role of oxysterols in AD pathogenesis [27]. In this study, we found 27-OHC induced significant morphological damage in the brain and impaired learning and memory ability, while 24S-OHC treatment caused the opposite effects. The treatment of 27-OHC and 24S-OHC disrupted the homeostasis in the serum and brain and changed the expression of the related metabolic enzymes (CYP27A1, CYP7B1, CYP46A1). Given that oxysterols could function as the ligands of RORγ to regulate its transcriptional activity and function, we measured the expression of transcription factors (RORγt, Foxp3) and the related immune factors (IL-17A, GM-CSF, MIP-3α, IL-10, IFN-λ2). Our results indicated that 27-OHC was able to promote the immune response but 24S-OHC inhibited abnormal immunity and showed a neuroprotective effect, which partly alleviated the neurotoxicity of 27-OHC.

Studies indicated that brain 27-OHC and 24S-OHC could be valuable surrogate markers in neurodegenerative diseases including AD. The proportion between 27-OHC and 24S-OHC might be an initial step to control the brain Aβ level [28]. Our previous studies have shown that 27-OHC, as the most abundant cholesterol derivatives and with high BBB permeability, induced cognitive impairment by the mechanisms of neuroinflammation, microbiota dysbiosis, synaptic dysfunction, pyroptosis, etc. [29,30,31]. However, excess cholesterol synthesized daily in the central nervous system is balanced basically by converting to 24S-OHC, which is transported out of brain, and loss of brain 24S-OHC was a risk factor in progression of AD [32], so it is now well-established that 24S-OHC might play a role in counteracting neurodegeneration. In our study, 0.15 mg/kg body weight 24S-OHC was selected to explore the neuroprotective effect based on the circulation 24S-OHC level of healthy people under physiological conditions [33,34,35,36,37,38].

In the current study, the novel object recognition and Morris water maze test were carried out to evaluate the effects of oxysterols on the learning and memory ability of mice. 27-OHC treatment group mice performed worse in the Morris water maze test with significantly longer mean distance to platform and fewer number of platform-site crossovers, suggesting that mice in 27-OHC group exhibited evident latency to locate the platform. In the novel object recognition test, there was a decreasing trend of the NORI, FDI and TDI values in 27-OHC-treated mice, although no statistical difference. 24S-OHC-treated mice exhibited shorter escape latency on Day 5, indicating that the mice might form memories of the platform position through continuous training. Moreover, mice in this group showed shorter distance to platform and increased numbers of platform-site crossovers, showing better memory retention of the platform location. Although no significant difference was observed in the 27-OHC+24S-OHC group, it still suggested that mice treated with the both oxysterols performed better than mice in the 27-OHC group. Meanwhile, 24S-OHC mitigated the brain morphologic injuries compared with the 27-OHC group.

Consistent with our previous study [39], treatment with 27-OHC caused significantly higher 27-OHC and lower 24S-OHC level in serum, while treatment with 24S-OHC led to higher 24S-OHC and lower 27-OHC level in serum. Accordingly, significant upregulation of CYP27A1 and downregulation of CYP46A1 were observed in the 27-OHC group; but in the 24S-OHC group, the opposite changes were observed in the expression of CYP27A1 and CYP46A1 compared to those of 27-OHC-treated mice; these results proved the treatment worked well and strengthened the regulation of these enzymes’ expression under treatment of 27-OHC and 24S-OHC. In 2016, Testa and collaborators reported that in the brain of AD patients, 24S-OHC was the only detected oxysterol with gradually decreasing concentration during the progression of AD. Simultaneously, increased CYP27A1 as well as decreased CYP46A1 were observed and the increased 27-OHC/24S-OHC ratio was consistent with AD progression [40]. The level of serum 24S-OHC in CYP27A1-overexpression mice was significantly reduced [41], which underlined the extensive involvement of CYP27A1 in the metabolism and homeostasis of oxysterols. In addition, the expression of CYP7B1 in 27-OHC group was significantly upregulated compared with the other three groups. Consistent with the present results, the protein expression of brain CYP7B1 was significantly downregulated in SD rats treated with inhibitor of CYP27A1 (Anastrozole) in our previous study [42]. CYP7B1 is a neuronal enzyme involved in 27-OHC catabolism, decreased activity of which was shown in the case of hypercholesterolemia, oxidative stress, inflammation, etc., to be a possible explanation for the increased 27-OHC [43,44]. Yau et al. reported that lower expression of CYP7B1 mRNA was also found in the dentate neurons of AD subjects [45]. The discrepancy might be due to that excess 27-OHC leading to the increase of CYP7B1 expression to accelerate its clearance and maintain the oxysterol homeostasis in the brain, but fails to reverse the changes of 27-OHC. Above all, it is clear that 27-OHC and 24S-OHC are signaling molecules with important implications for AD. Given that multiple oxysterols are involved in brain cholesterol metabolism and influence the progression of AD, more types of oxysterols should be studied in our future study.

The adaptive immune responses mediated by T lymphocytes is identified as a key player in AD etiology and pathogenesis [46]. As important subsets of CD4^+^ T lymphocytes, Th17 cells are highly immunogenic while Tregs cells play essential roles in maintaining immunological homeostasis [47]. The disruption of Th17/Treg balance in AD deserves further investigation. Our previous study indicated that the IL-17A level in the brain and plasma was highly increased in C57BL/6J mice treated with 27-OHC [31]. In this study, we found that 27-OHC burden induced RORγt mRNA and protein level in the brain, but treatment with 24S-OHC reduced RORγt protein level. Soroosh et al. reported that the hydroxyl group at the 27th carbon of oxysterols was a prerequisite for RORγt agonism and 27-OHC was proved to be one of the most potent and efficacious endogenous ligands by directly binding to the RORγ ligand-binding domain. Additionally, CYP27A1-deficient mice showed a significant reduction of Th17 cells, which was similar with the changes observed in RORγt-knockout mice [18], supporting the RORγt-dependent immune modulation of 27-OHC and the direct role of CYP27A1. Wang et al. demonstrated that 24S-OHC dose-dependent suppression of transcriptional activity and effectively competed for radioligand [3H]-25-hydroxycholesterol binding to RORγt [48]. Similarly, we found that the gene and protein expression level of RORγt were lower in 27-OHC+24S-OHC mice than in the 27-OHC group. Our study presents evidence that oxysterols are involved in Th17 cell-related signaling pathways by regulating the expression of RORγt in vivo. On the other hand, the immunosuppressive activity induced by Treg cells is important to the development and maintenance of AD, and Treg administration attenuates AD progression in AD model mice [49]. In the present study, the expression of Foxp3 in 24S-OHC group was significantly higher than the other three groups, and the trend was opposite to that of RORγt.

After establishing the association between oxysterols and immune response, we next investigated the changes of multiple factors regulating the Th17/Treg balance. As expected, IL-17A expression level was significantly increased in 27-OHC group compared with Control group, while both gene and protein level of IL-10 were downregulated. It’s notable that higher expressions of IL-17A, GM-CSF and MIP-3α were observed in 27-OHC group contrast to 24S-OHC mice, accompanied by lower expressions of IL-10 and IFN-λ2. In rats models of AD induced by injecting Aβ1-42 into the ventricle, Th17 cells were able to infiltrate into brain parenchyma through disrupted BBB and release the cytokines such as IL-17A, IL-22 and IL-23 in the blood and brain, the expression of brain RORγt was also upregulated, while the Treg-related cytokines such as IL-10 and TGF-β were decreased [14,50]. Zhang et al. reported an evident imbalance in the Th17/Treg ratios in the blood and brain samples of LPS-induced periodontitis mice, together with the activation of transcription cofactor STAT3 and impairment of learning and memory ability. The expressions of Th17-related cytokines (IL-17A, IL-21, IL-22 and IL-1β) were increased while that of Treg-related cytokines (IL-10 and IL-2) were decreased [51]. These results suggest that Th17/Treg balance and the related immune responses may be associated with the occurrence and progression of cognitive impairment. Additionally, it is well-characterized that highly pathogenic cytokines of Th17 cells like GM-CSF and MIP-3α were also found in AD patients and animal models [52,53,54]. While IFN-λ2 could reduce the generation of inflammatory cytokines in astrocytes and microglia cells, it will inhibit excessive infiltration of inflammatory cells into the brain and suppress the opening of the BBB [55]. These results are consistent with the present study, Th17/Treg imbalance and the dysfunction of related immune factors are associated with the oxysterol-induced changes of learning and memory ability. In the meantime, changes of IL-17A, GM-CSF (although no significant difference), IL-10 protein and IFN-λ2 gene expression supported the concept that 24S-OHC might play a part in alleviating the excessive or inappropriate immune responses induced by 27-OHC.

## 4. Materials and Methods

### 4.1. Animals and Treatments

Forty 9-month-old male C57BL/6J mice were purchased from SPF biotechnology co., LTD. (Beijing, China). All animals were kept in the animal house of Capital Medical University with 12 h light/dark cycle, humidity (50–60%) and temperature (22–25 °C) with standard diet and water ad libitum. The study was approved by the ethics committee of Capital Medical University (Ethics: AEEI-2014-047). The mice were randomly divided into 4 groups (10 in each group): Control group (saline 0.2 mL/d), 27-OHC group (5.5 mg/kg, i.p. daily), 24S-OHC group (0.15 mg/kg, i.p. daily) and 27-OHC + 24S-OHC group (5.5 mg/kg 27-OHC + 0.15 mg/kg 24S-OHC, i.p. daily). All mice received a daily subcutaneous injection for consecutive 21 days. Novel object recognition and Morris water maze test were conducted. Then, mice were anesthetized and blood samples and fresh tissues were collected, weighed and frozen at −80 °C until use.

### 4.2. Neurobehavioral Tests

#### 4.2.1. Novel Object Recognition Test

Novel object recognition test was performed to determine the short-term/working memory of mice as described in a previous study [56]. The mice were placed in an experimental box for object recognition training and testing. In brief, during the training period, two identical objects A1 and A2 were used as the familiar objects, being placed at two corners of testing box with equal distance from the wall. The mice were put into the box to explore the objects for 10 min. The test was conducted one hour later and A1 was replaced with B. The mice were released from the same position to explore the objects for another 10 min. Experimental box and objects were cleaned with 75% ethanol solution to remove odor cues after each operation. The NORI, FDI and TDI value were calculated as follows: NORI = Fn/Tn × 100%, FDI = (Fn − Ff)/(Fn + Ff), TDI = (Tn − Tf)/(Tn + Tf) (Tn: Total exploring frequency; Fn/Ff: Exploring frequency of novel/ familiar object; Tn/Tf: Exploring time of novel/familiar object).

#### 4.2.2. Morris Water Maze Test

The Morris water maze test was conducted to evaluate the spatial learning and memory of mice [42]. Briefly, a pool was filled with water (temperature 21 ± 1 °C) and titanium dioxide powder was added to improve the tracking ability of mice. An escape platform (10 cm in diameter, 38 cm in height) was placed at a constant position in the middle of southeast quadrant. During the five-day orientation navigation tests, animals were allowed to swim for 90 s to search for the platform, and they would be guided to the platform and stay for 15 s if they did not find it in 90 s. The probe trails were carried out with the platform removed on the sixth day. The mice were released from the northwest quadrant and allowed to swim for 90 s. The results were recorded individually for each mouse.

### 4.3. Hematoxylin-Eosin (HE) Staining

The whole brain was fixed with 4% paraformaldehyde and embedded in paraffin. Sections with proximately 5 μm in thickness were sequentially dewaxed, staining with Hematoxylin solution for 3–5 min and Eosin dye for 5 min, dehydrated and finally sealed with neutral gum.

### 4.4. High-Performance Liquid Chromatography-Mass Spectrometry (HPLC-MS)

The serum and brain levels of 27-OHC and 24S-OHC were detected by HPLC-MS according to our previous study [31,39]. In short, 50 μL serum or brain homogenate was prepared with 50 μL d5-27-OHC and d7-24S-OHC mixture as internal standard in it. The derivatization was performed with niacin, 4-Dimethylaminopyridine, N,N′-Diisopropylcarbodiimide and Dichloromethane mixtures and then we evaporated the contents in a vacuum dryer. The samples were redissolved in 100 μL methanol and centrifugated at 13,000 rpm for 10 min, after which the supernatants were collected in glass sample vials and analyzed by HPLC-MS.

### 4.5. Quantitative Real-Time PCR (qRT-PCR)

Total mRNA was isolated from mouse brains by SV Total RNA Isolation system (Promega Corporation, Madison, WI, USA). The mRNA levels of amyloid precursor protein (APP), SAA, CYP27A1, CYP7B1, CYP46A1, RORγt, Foxp3, IL-17A, GM-CSF, MIP-3α, IL-10 and IFN-λ2 were measured by qRT-PCR. In short, 1.0 μg of total RNA was added into a First Strand cDNA Synthesis Kit (ThermoFisher Scientific, Waltham, MA, USA). The primers were designed specifically after retrieving from the NCBI database and detailed in Table 2. qRT-PCR reactions were performed using KAPA SYBR^®^ PCR Kit (Kapa Biosystems, Woburn, MA, USA) according to the manufacturer’s instruction. GAPDH was used as an internal reference for normalization in this procedure. All samples were tested in triplicate from three biological replicates at least.

### 4.6. Western Blot

Approximately 40 mg of brain tissue lysed in RIPA buffer including protease inhibitors were homogenized and centrifuged at 12,000 rpm for 5 min at 4 °C. The concentration of protein was detected by a BCA Protein Quantitative Kit (Dingguo Changsheng Biotechnology, Beijing, China). Equivalent amounts of protein samples (40 ng) were separated by 10% SDS-PAGE and transferred to polyvinylidene fluoride membranes (PVDF). The antibodies used were as follows: APP 1:1000 (Abcam, ab126732, Cambridge, UK), SAA 1:1000 (Abcam, ab199030, Cambridge, UK), CYP27A1 1:1000 (Abcam, ab126785, Cambridge, UK), CYP7B1 1:1000 (ABclonal, A17872, Wuhan, China), CYP46A1 1:2000 (Abcam, ab244241, Cambridge, UK), RORγt 1:2000 (Abcam, ab207082, Cambridge, UK), Foxp3 1:1000 (Abcam, ab215206, Cambridge, UK), IL-17A 1:3000 (Abcam, ab189377, Cambridge, UK), GM-CSF 1:1000 (Proteintech, 17762-1-AP, Chicago, IL, USA), MIP-3α 1:1000 (Abcam, ab106151, Cambridge, UK), IL-10 1:1000 (Abcam, ab189392, Cambridge, UK), IFN-λ2 0.1 µg/mL (R and D, AF4635, Minneapolis, MN, USA). The protein density was measured by Image System Fusion FX (Vilber Lourmat, Paris, France) and GAPDH was used as the reference for standardization.

### 4.7. Statistical Analyses

Data analysis was performed using SPSS 23.0 (SPSS, Inc., Chicago, IL, USA) and GraphPad Prism 8.0.1 software. The data were presented as mean ± SEM or median with range according to the sample distribution. For parametric data, significant differences among groups were evaluated with one-way ANOVA and post hoc comparisons were conducted by LSD-t OR Dunnett T3 test. Two-way ANOVA analysis was used for repeated measurement data in Morris water maze test. Kruskal–Wallis test was conducted for nonparametric data. All statistical tests were 2-sided and statistical differences were considered when *p* < 0.05.

## 5. Conclusions

In conclusion, our study indicates that oxysterols participate in the changes of learning and memory ability by regulating Th17/Treg balance-related immune responses. The results support the neuroprotective effect of 24S-OHC which partially reverse the brain damages caused by 27-OHC. Taken together, it would be of great clinical significance to inhibit the accumulation of 27-OHC but increase 24S-OHC level in the brain to prevent AD.

## Figures and Tables

**Figure 1 ijms-23-04370-f001:**
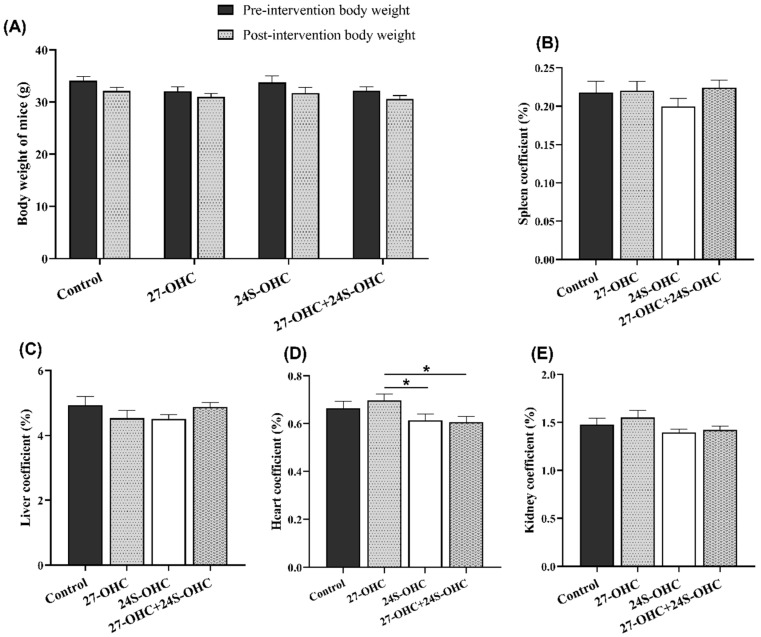
Effects of oxysterols on body weight and organ coefficient (*n* = 10, mean ± SEM). (**A**) Body weight of mice pre- and post-intervention; (**B**–**E**) organ coefficient of spleen, liver, heart and kidney (organ weight/body weight, %). *: *p* < 0.05.

**Figure 2 ijms-23-04370-f002:**
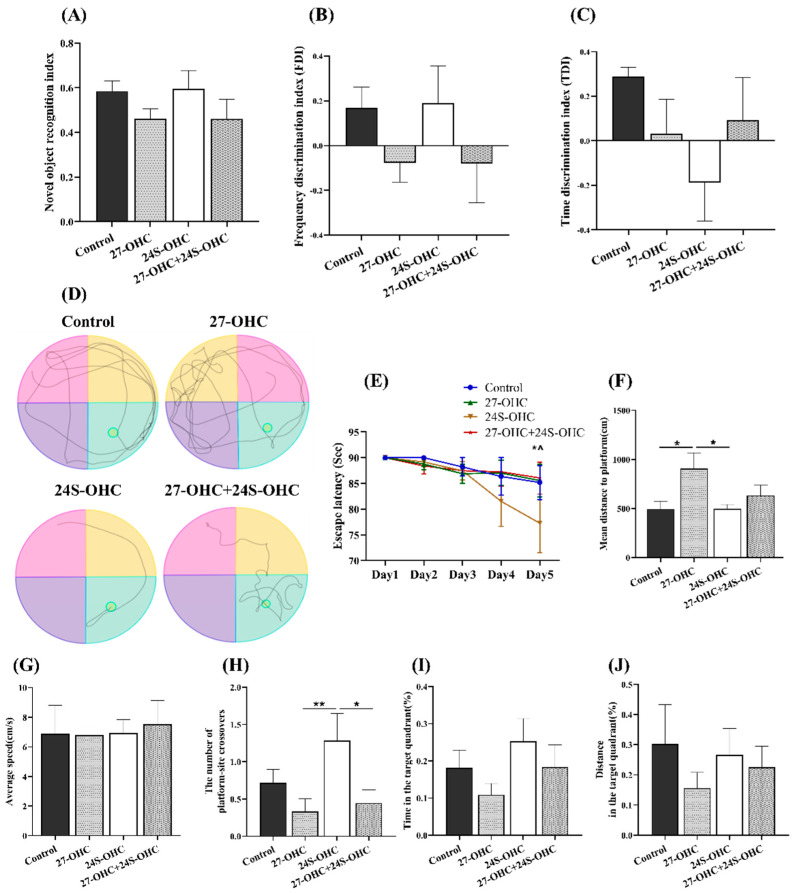
Effects of oxysterols on novel object recognition and Morris water maze test (*n* = 10, mean ± SEM). (**A**) Novel object recognition index (NORI); (**B**) frequency discrimination index (FDI); (**C**) time discrimination index (TDI); (**D**) swimming path; (**E**) escape latency, *: 24S-OHC group: Day1 VS Day5, *p* < 0.05, ^: 24S-OHC group: Day2 VS Day5, *p* < 0.05; (**F**) mean distance to platform; (**G**) average speed; (**H**) the number of platform-site crossovers; (**I**) time in the target quadrant (%); (**J**) distance in the target quadrant (%). *: *p* < 0.05, **: *p* < 0.01.

**Figure 3 ijms-23-04370-f003:**
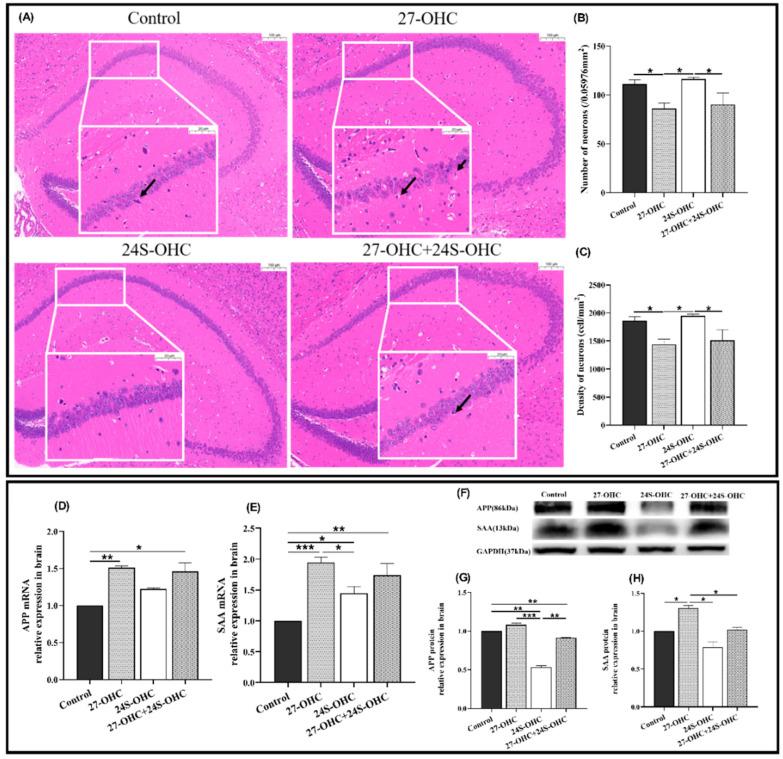
Effects of oxysterols on brain pathology and the expression of amyloid precursor protein in the brain. (**A**) HE staining of the whole brain (*n* = 3, scale bar: 100 and 20 μm), black arrows: Nuclei pyknosis; (**B**) number of neurons (/0.05976 mm^2^) (*n* = 3, mean ± SEM); (**C**) density of neurons (cell/mm^2^) (*n* = 3, mean ± SEM); (**D**) APP mRNA (*n* = 7, median with range); (**E**) SAA mRNA (*n* = 7, mean ± SEM); (**F**) Western blot results of APP and SAA; (**G**) APP protein (*n* = 7, mean ± SEM); (**H**) SAA protein (*n* = 7, mean ± SEM). *: *p* < 0.05, **: *p* < 0.01, ***: *p* < 0.001.

**Figure 4 ijms-23-04370-f004:**
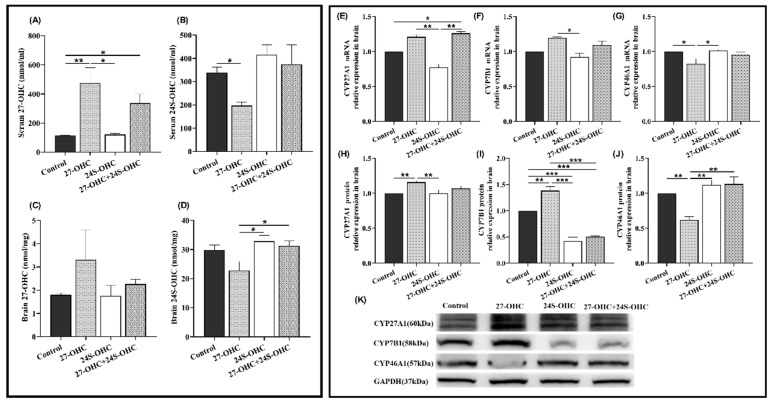
Changes of serum/brain oxysterols and the expression of metabolic enzymes in the brain (*n* = 7). (**A**) Serum 27-OHC (median with range); (**B**) serum 24S-OHC (mean ± SEM); (**C**) brain 27-OHC (mean ± SEM); (**D**) brain 24S-OHC (mean ± SEM); (**E**) CYP27A1 mRNA (mean ± SEM); (**F**) CYP7B1 mRNA (median with range); (**G**) CYP46A1 mRNA (mean ± SEM); (**H**) CYP27A1 protein (mean ± SEM); (**I**) CYP7B1 protein (mean ± SEM); (J) CYP46A1 protein (mean ± SEM); (**K**) Western blot results of CYP27A1, CYP7B1 and CYP46A1 *: *p* < 0.05, **: *p* < 0.01, ***: *p* < 0.001.

**Figure 5 ijms-23-04370-f005:**
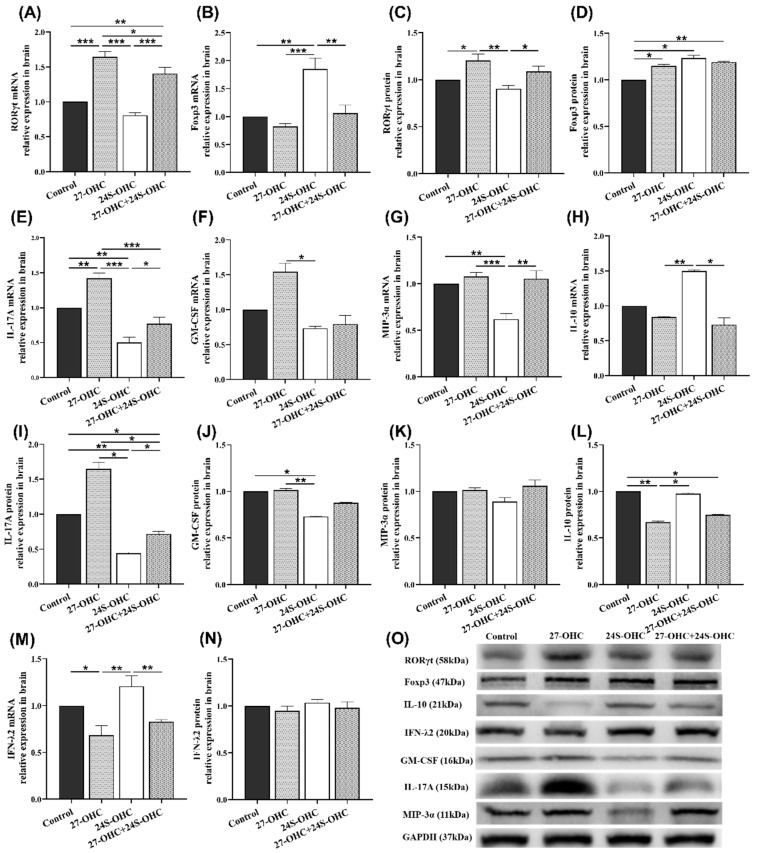
Effects of oxysterols on the expression of transcription and immunomodulatory factors of T cells in the brain (*n* = 7). (**A**) RORγt mRNA (mean ± SEM); (**B**) Foxp3 mRNA (mean ± SEM); (**C**) RORγt protein (mean ± SEM); (**D**) Foxp3 protein (mean ± SEM); (**E**) IL-17A mRNA (mean ± SEM); (**F**) GM-CSF mRNA (median with range); (**G**) MIP-3α mRNA (mean ± SEM); (**H**) IL-10 mRNA (median with range); (**I**) IL-17A protein (mean ± SEM); (**J**) GM-CSF protein (mean ± SEM); (**K**) MIP-3α protein (mean ± SEM); (**L**) IL-10 protein (mean ± SEM); (**M**) IFN-λ2 mRNA (mean ± SEM); (**N**) IFN-λ2 protein (mean ± SEM); (**O**) Western blot results of IL-17A, GM-CSF, MIP-3α, IL-10 and IFN-λ2. *: *p* < 0.05, **: *p* < 0.01, ***: *p* < 0.001.

**Table 1 ijms-23-04370-t001:** FDI and TDI values in novel object recognition test (*n* = 10).

Groups	FDI	TDI
Control group	0.17 ± 0.23	0.29 ± 0.09
27-OHC group	−0.08 ± 0.26	0.031 ± 0.46
24S-OHC group	0.19 ± 0.40	−0.19 ± 0.39
27-OHC+24S-OHC group	−0.08 ± 0.52	0.09 ± 0.51
F value	1.099	1.098
*p*-value	0.367	0.371

**Table 2 ijms-23-04370-t002:** Primers used in this study.

Primer	Forward Sequence (5′-3′)	Reverse Sequence (5′-3′)
APP	TGAATGTGCAGAATGGAAAGTG	AACTAGGCAACGGTAAGGAATC
SAA	ACACTGACATGAAGGAAGCTAA	CCTTTGAGCAGCATCATAGTTC
CYP27A1	ATCGCACAAGGAGAGCAATGGTAC	GGCAAGGTGGTAGAGAAGATGAGC
CYP7B1	AACCCTTTCCAGTACCAGTATG	GTGAACGTCTTCATTAAGGTCG
CYP46A1	CTTGGACATCTCCCCTACTTTT	TCAGGAACTTCTTGACTGACTC
RORγt	ACAAATTGAAGTGATCCCTTGC	GGAGTAGGCCACATTACACTG
Foxp3	TTTCACCTATGCCACCCTTATC	CATGCGAGTAAACCAATGGTAG
IL-17A	GAGCTTCATCTGTGTCTCTGAT	GCCAAGGGAGTTAAAGACTTTG
GM-CSF	TTCAAGAAGCTAACATGTGTGC	GGTAACTTGTGTTTCACAGTCC
MIP-3α	TCTTCCTTCCAGAGCTATTGTG	GACTGCTTGTCCTTCAATGATC
IL-10	TTCTTTCAAACAAAGGACCAGC	GCAACCCAAGTAACCCTTAAAG
IFN-λ2	GGATTGCCACATTGCTCAGTTCAAG	GTCCTTCTCAAGCAGCCTCTTCTC
GAPDH	GGTTGTCTCCTGCGACTTCA	TGGTCCAGGGTTTCTTACTCC

## Data Availability

The data presented in this study are available on request from the corresponding author.

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
