# Peer review of "Regulation of Th17/Treg Balance by 27-Hydroxycholesterol and 24S-Hydroxycholesterol Correlates with Learning and Memory Ability in Mice"

_ijms, 2022, doi:10.3390/ijms23084370_

Round 1

Reviewer 1 Report

The Ms. by Wang et al demonstrates that the balance between the oxysterols 27-OHC and 24S-OHC affect the learning and memory ability and the immune response in C57BL/6J mice. This is a relevant observation that may impact in our understanding of the etiology of Alzheimer’s disease. Nevertheless, the study has a number of caveats that need to be addressed before it can be considered for publication.

Major points

  1. There is no experimental evidence supporting the hypothesis that 27-OHC and 24S-OHC affect the learning and memory ability just by disturbing Th17/Treg balance. Therefore, the title should be accordingly modified.
  2. English should be revised by a native speaker.
  3. Line 88: An explanation of the design and purpose of the experiments that are described in this section should be included.
  4. There is no indication of the statistical tests that are used in the different experimental conditions (one-way ANOVA, two-way ANOVA, Student’s t test,…).
  5. Lines 126-129: The statements indicating that “Nuclei pyknosis and enlarged intercellular space were observed to varying degrees.” and “treatment with 27-OHC+24S-OHC partially alleviated the disordered arrangement and reduced numbers of neurons” should be supported by quantitative data.
  6. The Abeta and SAA immunohistochemistry shows non-specific nuclear staining (DAPI). In contrast of the author’s claim, Abeta deposition is never detected in the brain of C57BL6/J mice, further supporting that the immunolabeling is not specific. Results illustrated by Figure 3 should be removed from the study. The claims on the capacity of 27-OHC to induce severe Abeta deposition in the brain that are present in the discussion (line 221-222) should be removed as well.

Minor points

  1. Lines 54-56: references 11 and 12 are focused on Parkinson disease (not on Alzheimer’s disease).
  2. The Introduction should include information about Th17 and Treg cells and its relevance in neuroinflammation. CYP7B1 function should be described in the Introduction (or, at least, when it is mentioned in the Results section)
  3. What does “organ coefficient” means?
  4. Lettering of figures is too small. It cannot be read.
  5. Line 185: References should be provided to justify the use of RORgammat and Foxp3 as markers of Th17 and Treg cells.
  6. To unify the nomenclature, use either IL-17A or IL-17.

Reviewer 2 Report

Dysregulations in the metabolism of cholesterol and its oxidative metabolites oxysterols in the brain have been reported in several neurodegenerative diseases and also in brain aging. The oxysterols 27-hydroxycholesterol (27-OHC) and 24-hydroxycholesterol (24-OHC) seem to have a key role in the pathogenesis of Alzheimer’s disease (AD), according to several recent studies. Some of these studies turn the spotlight on the natural ligands of these oxysterols as bioactive molecules, being oxysterols established ligands to important nuclear receptors that act as transcription factors, in that regulating the expression of important proteins involved in the physiopathologic mechanisms of AD.

In this manuscript, the authors explored the hypothesis that the two oxysterols 27-OHC and 24-OHC may modulate the activity of one of those receptors, the retinoic acid-related orphan receptor gamma t (RORϒt), in turn resulting in a Th17 lymphocyte mediated and dysfunctional immune response. They used a murine model (mice treated with the oxysterols) and demonstrated that 27-OHC treated mice have learning and memory deficits, more Aβ deposition, changes in metabolic enzymes like CYP27A1, CYP7B1 and CYP46A1 in the brain and altered immunomodulatory factors in the brain, especially involved in Th17/Treg balance, accompanied by altered RORϒt and Fox3p expression.

The topic is intriguing and results seem promising, however several major revisions must be made prior to the publication of this manuscript.

Major points to be addressed:

1) English language must be thoroughly revised, several sentences are poorly expressed and there are several grammar errors.

2) Introduction:

- I suggest a brief explanation of cholesterol metabolism, how, where and when 27-OHC and 24-OHC are produced in the body.

- Similarly, I suggest a brief explanation of how oxysterols act on several nuclear receptors, then focusing the attention on 27-OHC and 24-OHC actions on RORϒt

- Again, the role of T lymphocytes, particularly of Th17 cells, in the pathogenesis of AD is not clearly explained. Please remodulate this part and deepen his point, that is crucial for the topic of the manuscript.

- The final sentence is too strong and should be mitigated, since the causes of AD are multiple while 27-OHC and 24-OHC may contribute to the pathogenesis. Further, authors did not demonstrate a Th17/Treg imbalance, but only the imbalance of immunomodulatory factors involved in functions of these cells. I suggest to rewrite this sentence, stressing the starting hypothesis and how the authors organized the study, postponing results and their discussion to the appropriate sections.

3) All figures must be reorganized, characters are completely unreadable

4) Results: in general the description of results are hard to follow especially for the inappropriate use of English language

- Also because methods are placed after the discussion, authors should briefly describe the study organization and treatments at the beginning of the Results section

- in section 2.1 the authors showed that heart coefficient in 27-OHC group is different in respect to 24-OHC and 27-OHC+24-OHC groups, but not versus control group. How the authors explain this observation?

-in section 2.2 the abbreviations FDI etc are not clearly explained, please expand the abbreviations

- the authors say that 24-OHC increased TDI but in Table 1 the value is negative, so decreased?

- how the authors measured mice learning and memory ability? The explanation is not clear both in results and methods, especially for Morris water maze test. Please deepen this explanation

- figure 3A is not clear, I am not an expert in histopathology but I did not see differences in the 4 figures of figure 3A. Perhaps a higher magnification could help

-figure 5O: the expression of RORϒt does not seem upregulated in 27-OHC treated mice. I suggest to use a better image

5) Discussion: is not clear in several points, mostly because of the inappropriate use of terms

- lane 229: the term immunogenic is inappropriate

-lane 257: the term “alleviated the results” is inappropriate

-lanes 268-273 are not clear, please rewrite the sentence

-lanes 274 is too ambitious: the pathogenic mechanisms of AD are several others, imbalance between 27-OHC and 24-OHC levels may contribute

-lanes 295-297: the induction of a receptor by a ligand does not mean that the receptor is activated. Please deepen this point.

6) Materials and Methods:

- please explain why this dose of oxysterols was used to treat animals

- as stated above, please better explain how neurobehavioral measures were obtained

Round 2

Reviewer 1 Report

I still have two concerns:

1. There is no direct evidence that the Th17/Treg imbalance causes the cognitive deficits observed in the experiments described in this manuscript. Other hypothetical mechanisms may be participating, thus invalidating the claim that “27-hydroxycholesterol and 24S-hydroxycholesterol affect the learning and memory ability by regulating Th17/Treg balance-related immune responses”. A loss-of-function experiment demonstrating that reversal of the Th17/Treg imbalance results in the improvement of learning and memory is missing. In the absence of this evidence the title should be modified accordingly (for instance, “Regulation of Th17/Treg balance by 27-hydroxycholesterol and 24S-hydroxycholesterol correlates with learning and memory ability in mice”). The paper by Zhang et al (2021) does not includes direct evidence that the inhibitor of the STAT3 signaling pathway can recover the cognitive impairment induced by LPS-induced periodontitis in mice. Actually, the last sentence of the results section of Zhang et al (2021) states “This indicated that cognitive impairment, accompanied by increase of LPS inside and outside the brain, MAY BE associated with the STAT3 pathway”. The other papers mentioned by the authors provide just circumstancial evidence for their claim.

2. The Abeta labeling claimed to be specific fully coincides with neuronal nuclei. Abeta deposits are extracellular, they are of size greater than that of cell nuclei, and they also localize within the neuropil. Therefore, the staining shown in Figure 3 is just an artifact and, therefore, this figure must be removed. [By the way, figure 3 has a number of defects, uses G instead of J, and the legend does not fit with the figure.]

Reviewer 2 Report

The authors have adequately replied to the comments.

Author Response

Dear reviewers,

Thank you for your comments on our manuscript entitled “27-hydroxycholesterol and 24S-hydroxycholesterol affect the learning and memory ability by regulating Th17/Treg balance-related immune responses” (ID: ijms-1648648). Those comments are very valuable and helpful for revising and improving our paper. We would be glad to respond to any further questions and comments that you may have.